## RESEARCH ARTICLE

# Dynamic expression and differential requirement of the myocyte fusogen Myomixer during distinct myogenic episodes in the zebrafish

Sunandan Dhar[1,2,3,‡], Serena Thomas[1,3,‡], Hui Li Yeo[3,*], Timothy E. Saunders[1,2,3,§] and Sudipto Roy[3,4,§]

## ABSTRACT

Skeletal muscle formation involves the fusion of myocytes into precisely aligned, multinucleated myofibres. These fibres continue to grow through reiterative rounds of myocyte fusion, incorporating new myonuclei and supporting muscle growth, repair and regeneration over organismal life span. The vertebrate-specific myocyte fusogens, Myomaker (Mymk) and Myomixer (Mymx), are crucial for generating multinucleated skeletal muscles. Here, using quantitative imaging and a *mymx* knockout strain, we explored the impact on myogenesis at different life stages of the zebrafish. We demonstrate that during the initial phase of muscle formation, *mymx* has a spatiotemporally varied expression across all axes of the developing myotome, not just along the anterior-posterior axis. On Mymx loss, myotome morphogenesis is disrupted, with both cell and tissue structure impacted. In particular, the shape of the resulting myotome segments is altered. Moreover, we show differential effects of Mymk versus Mymx loss on myocyte fusion and muscle growth. Finally, we report that perturbation to adult muscle multinucleation and size impacted bone development, again with different phenotypic severities among the two fusogen mutants. Together, our work provides insights into the interplay between myocyte fusion, myotome morphogenesis and acquisition of final adult form.

KEY WORDS: Myogenesis, Myocyte, Fusogen, Myomaker, Myomixer, Zebrafish, Skeleton, Adult form

## INTRODUCTION

During skeletal myogenesis, post-mitotic muscle precursors, the myocytes, undergo fusion to generate multinucleated myotubes, which assemble a macromolecular complex of contractile proteins and mature into precisely aligned striated muscle fibres (Wherley et al., 2024). This process involves a coordination of genetic factors and signals from the *in vivo* tissue environment. Proper

[1]Warwick Medical School, Division of Biomedical Sciences, University of Warwick, CV4 7AL Coventry, United Kingdom. [2]Mechanobiology Institute, National University of Singapore, Singapore 117411. [3]Institute for Molecular and Cell Biology (IMCB), Neurometabolism Division, Agency for Science, Technology and Research (A*STAR), 61 Biopolis Drive, Singapore 138673. [4]Department of Pediatrics, Yong Loo Lin School of Medicine, National University of Singapore, 1E Kent Ridge Road, Singapore, 119288, Singapore.
*Present address: Mandai Nature, 80 Mandai Lake Road, Singapore 729826.
[‡]These authors contributed equally to this work

[§]Authors for correspondence (timothy.saunders@warwick.ac.uk; sudipto_roy@a-star.edu.sg)

T.E.S., 0000-0001-5755-0060

morphogenesis of skeletal myofibres is crucial for force generation, necessary for moving body parts as well as the whole organism. Failure of skeletal muscle to form correctly results in a plethora of human diseases, including muscular dystrophies, dystrophinopathies, sarcoglycanopathy and integrin-related diseases (Barresi et al., 2000; Hayashi et al., 1998; Ortiz-Lopez et al., 1997). In addition, there are myotubular and centronuclear myopathies (Amburgey et al., 2017; Tasfaout et al., 2018), which are directly related to defects in early muscle morphogenesis. Impaired myocyte fusion and improper morphogenesis of myofibres can also lead to developmental myopathies such as Carey-Fineman-Ziter syndrome (Di Gioia et al., 2017). Zebrafish has been at the forefront of understanding muscle development and muscle disease (Bassett and Currie, 2003; Lieschke and Currie, 2007; Dalle Carbonare et al., 2025), and is highly suitable for testing the efficacy of translationally relevant drugs (Karuppasamy et al., 2024). Importantly, understanding muscle development can reveal insights into how tissues repair, regenerate (Oudhoff et al., 2024; Gemberling et al., 2013; Kaliya-Perumal and Ingham, 2022) and age (Quach et al., 2024), which are highly pertinent for long-term health of an aging population.

Skeletal muscle typically consists of fast and slow-twitch fibres (Miao and Pourquie, 2024). In the zebrafish, myocytes committed to the fast-twitch fate undergo fusion as the myotome forms, whereas the slow fibres remain mononucleated initially, becoming multinucleated during post-embryonic development (Kimmel and Warga, 1987; Devoto et al., 1996; Currie and Ingham, 1998; Hughes and Salinas, 1999; Roy et al., 2001). The fusion-specific proteins, Mymk (Millay et al., 2013) and Mymx (also known as Minion and Myomerger) (Bi et al., 2017; Quinn et al., 2017; Zhang et al., 2017), are crucial for myocyte fusion across vertebrate species (Luo et al., 2015; Zhang and Roy, 2017). Mymk is a transmembrane protein that enables hemifusion between the cell membranes of fusing myocytes. The structure and functional domains of Mymk have been well characterised (Millay et al., 2016). On the other hand, the micropeptide Mymx, remains less well studied. It is known to act independently of Mymk, generating local membrane tension to promote fusion pore formation at the sites of hemifusion (Leikina et al., 2018). The *in silico* predicted secondary structure of Mymx contains two α-helices, which are thought to be essential for its catalytic activity (Gamage et al., 2022). In the mouse, Mymk and Mymx are also required for muscle regeneration; their absence in muscle stem cells, the satellite cells, abrogates myocyte fusion post-injury (Bi et al., 2018).

In the zebrafish embryo, both *mymk* and *mymx* mRNA have been reported to be expressed in the myotome during early myogenesis (Landemaine et al., 2014; Shi et al., 2017), starting a few hours after segmentation of the somites and continuing throughout myotome morphogenesis. By 30 h post-fertilisation (hpf), only the youngest,

posterior-most segments express *mymk* and *mymx*, as the more mature myotomes have already completed myogenesis. However, these studies have only revealed the expression of the fusogens at the whole-embryo scale. To better understand their expression at a cellular-scale and at different stages of myotome formation, a higher spatial resolution is required. We have previously reported spatiotemporal patterns of *mymk* expression at the cellular-scale using fluorescence *in situ* hybridisation (Mendieta-Serrano et al., 2022). However, a similar quantitative analysis of *mymx* expression is currently lacking, especially in 3D.

In addition to their native expression, it is important to also understand the functional importance of these fusogens during muscle formation. *mymk*$^{-/-}$ zebrafish embryos have been reported to differentiate mononucleated fast-twitch myofibres (Zhang and Roy, 2017) and have impaired myocyte fusion throughout the life of the organism (Shi et al., 2018). This is in contrast to the cell adhesion proteins, Jamb and Jamc, which are required for embryonic myocyte fusion (Powell and Wright, 2011) but are dispensable during fusion at post-embryonic stages (Si et al., 2019). Similarly, in homozygous *mymx*$^{-/-}$ embryos there is a disruption in myocyte fusion during embryonic myogenesis (Shi et al., 2017). Mymx also has an important role in larval muscle development, since there is an excessive adipose infiltration in mutant muscle fibres in homozygous *mymx*$^{-/-}$ adults (Wu et al., 2022). These fish showed reduced body size and a reduction in swimming ability. However, it remains unclear how loss of *mymx* impacts muscle morphogenesis at cellular and tissue scales. Furthermore, other developmental processes, such as bone formation, also depend on the generation of forces by the musculature (Felsenthal and Zelzer, 2017; Nowlan et al., 2010). How reduced muscle contractile efficiency impacts skeleton formation remains largely unexplored, since most mouse and zebrafish mutants with muscle differentiation defects are embryonic lethal. These are crucial questions, especially in light of the fact that mutations in the human *MYMX* gene result in developmental defects affecting muscle tissues throughout the life of the affected individuals (Ramirez-Martinez et al., 2022).

Here, we focus on three questions: (1) what is the full 3D expression pattern of *mymx* in the early embryo, during the first wave of myocyte fusion; (2) how are cell and tissue architectures altered upon loss of Mymx function; and (3) how does loss of Mymx function impact skeleton formation in the adult? In the first part of this study, we provide a 3D cell-scale analysis of *mymx* expression during the initial stages of myotome formation in the zebrafish embryo. We next explored the functional role of Mymx by generating a stable knockout line and show quantitatively that the absence of Mymx disrupted the dynamics of myotome morphogenesis and impacted the structure of developing myofibres. Intriguingly, defects in muscle formation on loss of *mymx* function were complex, with distinct differences at cell and tissue scale. Furthermore, we found that impaired muscle differentiation due to defects in myocyte fusion caused significant defects in skeletal morphogenesis, uncovering how the musculature can influence development of a properly patterned skeleton and acquisition of the adult form.

## RESULTS

### Dynamics of *myomixer* expression during embryonic muscle formation

*mymx* is known to be expressed in the developing zebrafish myotome soon after somite segmentation (Shi et al., 2017). However, to understand the role of *mymx* expression at different stages of myotome formation, a higher spatial resolution is required. We used fluorescence *in situ* hybridisation to resolve patterns in the

expression of *mymx* mRNA within individual myotomes at 24 hpf, where we have previously documented frequent fusion events (Mendieta-Serrano et al., 2022). As somite segmentation proceeds in an anterior-to-posterior wave over time, we were able to study temporal patterns (with a 30-min time resolution) in *mymx* expression by observing somites at different developmental stages during myofibre morphogenesis.

*mymx* mRNA appeared to be expressed evenly within cells in the myotome during somite stages S8 to S16, where muscle fibre morphogenesis is underway (Fig. 1A). Cross-sectional view of the myotomes revealed that the expression was limited to certain regions along the mediolateral (ML) axis, and this region shifted with different somite stages (Fig. 1B). We quantified *mymx* expression levels by measuring fluorescence intensities of the *in situ* signal in myotomes from three embryos. We observed *mymx* expression increased with myotome maturity in somite stages S7 to S12, before decreasing again in older somites (Fig. 1C). The somite stages when *mymx* expression was at its highest, S10 to S14, corresponded to the time when the highest number of myocyte fusion events have been reported to occur (Mendieta-Serrano et al., 2022).

To identify spatiotemporal patterns in finer detail, we analysed how *mymx* expression along specific embryonic axes changed during the key stages of myofibre morphogenesis (stages S8 to S16) (Fig. 1D-F). Along the anteroposterior (AP) axis, *mymx* was at its highest in the S12 stage, though the distribution was noisy (Fig. 1D); there were clear peaks in expression, but no clear spatiotemporal patterns when comparing the different somite stages. On the other hand, along the dorsoventral (DV) axis, there were two expression peaks positioned on either side of the DV midline (Fig. 1E). This was expected, since the cells at the DV midline of the myotome develop into muscle pioneers (Currie and Ingham, 1996), which are mononucleated slow-twitch muscles and typically do not express *mymx*. Low levels of *mymx* expression have been qualitatively observed in some slow muscle precursors (adaxial cells), although whether this expression has any functional impact is unknown (Yong et al., 2024). *mymx* expression was lowest in the dorsal- and ventral-most regions of the myotomes, which are thinner and contain fewer cells.

Along the ML axis, the peak of *mymx* expression shifted from medial to lateral positions with increasing myotome age (Fig. 1F). In early myotome segments (stage S8), *mymx* was expressed only in the layer of cells adjacent to the notochord and neural tube (Fig. 1B,F). As the myotome developed, *mymx* was expressed in layers of cells at increasing lateral distances, until it was only expressed in cells near the superficial edge of the myotome. This wave of *mymx* expression correlated closely with the previously reported wave of cell-cell fusion along the ML-axis (Mendieta-Serrano et al., 2022).

Thus, *mymx* has a complicated pattern of expression at 24 hpf, not limited to the AP axis. Along the ML-axis, peak in expression shifted from medial to lateral domains as the segments matured. Along the DV-axis, expression peaked on either side of the DV midline. Thus, like *mymk* (Mendieta-Serrano et al., 2022), *mymx* has a double-peaked expression pattern along the DV axis and a wave of expression from medial-to-lateral domains of the developing myotome. While the expression pattern closely correlates with the pattern of fusion events, it remains unknown what molecular mechanisms control this dynamic expression pattern.

### A *mymx* loss-of-function mutation inhibited embryonic myocyte fusion

To investigate the functional role of Mymx in skeletal muscle formation, we generated a stable mutant line with a 28 bp deletion in

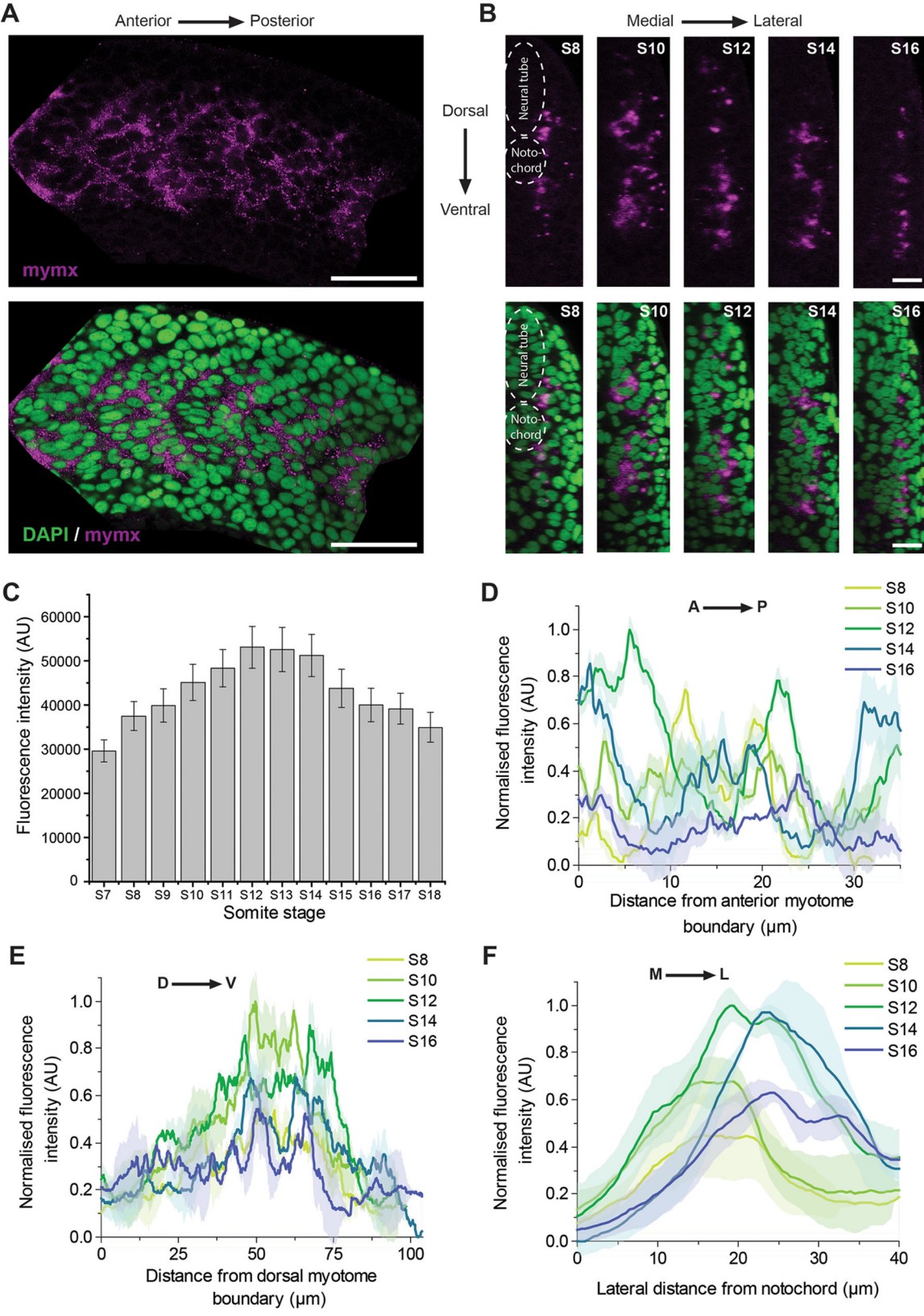

**Fig. 1. Spatiotemporal pattern of *mymx* mRNA expression.** (A) Fluorescence *in situ* hybridisation for *mymx* mRNA expression (magenta, top panel) in wild-type embryos fixed at 24 hpf co-stained with DAPI (green, bottom panel) (scale bars: 50 µm). (B) Transverse sections of myotomes showing *mymx* expression at different somite stages (scale bars: 20 µm). (C) Quantification of fluorescence intensities of *mymx* expression at different somite stages. Error bars represent the standard deviation (*n*=3 embryos). (D-F) Spatial distribution of *mymx* expression in myotomes at different somite stages along the anteroposterior (D), dorsoventral (E), and mediolateral (F) axes. The solid lines represent the mean and shaded areas represent the standard deviation (*n*=3 embryos).

the *mymx* coding region (see the Materials and Methods, Fig. 2A). This caused a frameshift mutation, predicted to disrupt both α-helices at the C-terminal tail of the protein (Fig. 2A). This is likely to result in a complete loss-of-function condition for the Mymx protein, similar to the effect of previously published alleles, since both of these helices are required for the fusogenic activity of Mymx (Gamage et al., 2022) (Fig. S1). Previous *mymx* mutants in zebrafish led to premature stop codons, effectively removing the extracellular domain entirely. Likewise, the peptide encoded by our *mymx* mutant allele is predicted to possess a grossly mutant extracellular domain, which features a complete loss of the highly conserved AxLyCxL motif that is considered essential for myocyte fusion (Fig. 2A and Fig. S1).

*mymx*$^{-/-}$ embryos were fixed at different stages of development and immunostained to study the morphology of skeletal myofibres. We observed that fast myofibres were mononucleated at 30 and 48 hpf, while those in wild-type siblings were multinucleated (Fig. 2B, top and middle rows). Myofibre elongation and alignment did not appear to be affected in the mutant embryos, similar to *mymk*$^{-/-}$ embryos previously reported (Zhang and Roy, 2017).

At 30 hpf in wild-type embryos, by which time the first wave of trunk muscle formation is complete, multiple nuclei in each muscle cell are distributed rather evenly along the length of the fibres (Fig. 2B, top row). However, at this stage in *mymx*$^{-/-}$ embryos we observed that the single nucleus of each mononucleate fibre was located towards the centre of the muscle fibres (Fig. 2B, top row). By 48 hpf, the nuclei were more evenly packed in a chevron-shaped pattern within the somite, aligning along the centre of the muscle fibres in *mymx*$^{-/-}$ embryos (Fig. 2B, middle row). During the larval stage (120 hpf), a subset of *mymx*$^{-/-}$ myofibres appeared multinucleated. However, the proportion of such cells was much less than in the wild-type siblings (Fig. 2B, bottom). Notably, the organisation of the nuclei was less regular than the wild-type, with nuclei in the mutant muscles clustering more, with less even distribution along the fibres.

We next quantified the number of nuclei in each myofibre from somites 9-20 to gain a more detailed understanding of how the loss of *mymx* function impacted myocyte fusion (Fig. 3A). Since myotome maturation and myofibre morphogenesis proceeds in an anterior-to-posterior wave, myotomes were grouped based on somite numbers and from three embryos for each group. Wild-type myofibres were mostly bi- and tri-nucleated, with a higher degree of multinucleation in the more mature myotomes (somites 9-12) relative to the younger ones (somites 13-16 and 17-20) (Fig. 3A). *mymx*$^{-/-}$ myofibres were almost entirely mononucleated at the 30 and 48 hpf stages, and only around 1% of myofibres appeared binucleated in somites 9-12. However, by the 120 hpf stage, around 5% of *mymx*$^{-/-}$ myofibres were binucleated in all of the somites studied (Fig. 3A, right). This indicates that a low level of myocyte fusion can occur at the larval stage despite the absence of Mymx activity. An alternative explanation for binucleated cells could be that cells are undergoing division. Although skeletal muscle precursors can undergo division (Borowik et al., 2023, 2024), we only found mononucleated cells at this stage in the *mymx*$^{-/-}$ loss-of-function embryos.

### Absence of *mymx* altered myofibre morphology

We next examined whether the lack of myocyte fusion in *mymx*$^{-/-}$ embryos impacted muscle tissue structure. We first quantified the aspect ratio of myofibres in wild-type and mutants at different embryonic stages (Fig. 3B). Myofibres in *mymx*$^{-/-}$ embryos had a significantly higher aspect ratio at 30 and 48 hpf. Mutant myofibres were elongated along the anteroposterior axis similar to the

wild-type but were considerably thinner (Figs 2B and 3B), resulting in higher aspect ratios. In wild-type embryos, multiple myocytes contribute their cytoplasm and cell membranes to myofibres spanning the AP extent of the myotome. In contrast, in *mymx*$^{-/-}$ embryos, a single myocyte needs to elongate and stretch out across the same width. However, the aspect ratios became similar for wild-type and mutant at the 120 hpf larval stage, with only the younger somites (S17-S20) showing a small difference (Fig. 3B, right). This reveals that loss of *mymx* function impacts cell morphology during embryonic stages, especially earlier in development. This is consistent with recent work that has quantified how the process of fusion is important in shaping the first formed muscle fibres of the zebrafish embryo (Mendieta-Serrano et al., 2025 preprint).

### *mymx* disruption affected myotome structure at both embryonic and larval stages

Do differences in myofibre morphology result in structural changes at the whole-tissue scale? We next compared the overall structure of the myotomes in wild-type and *mymx*$^{-/-}$ embryos. We found the angles of the chevron-shape of the myotomes were slightly higher in the mutant embryos at 30 hpf than in the wild-type (Fig. 4A). However, this was reversed in 48 hpf and 120 hpf myotomes, where the chevron angle was lower in mutants. These altered angles may be due to the differences in elongation and rearrangement dynamics of *mymx*$^{-/-}$ mononucleated fast-twitch muscles within the developing myotomes; both of these factors play a role in formation of the chevron (Tlili et al., 2019). However, these differences were less apparent in younger somites (S17-S20) at all stages.

The overall size of myotome segments was also altered in the mutants (Fig. 4B,C). Myotomes in *mymx*$^{-/-}$ embryos were expanded in terms of their dorsoventral length (Fig. 4B) as well as whole-myotome volumes (Fig. 4C) at the 30 and 48 hpf stages. While the difference was less pronounced for younger somites at 30 hpf, there was a clear difference for all myotomes at 48 hpf. It is possible that the larger myotome size is necessitated by a higher number of myofibres present in the mutant myotomes from the lack of fusion compared to the wild-type (Fig. 4D). Both mutant and wild-type had a similar number of nuclei per myotome at all stages (Fig. S2A,B), but the lack of fusion in *mymx*$^{-/-}$ mutants meant that typically each nucleus represented an individual elongated myofibre.

Interestingly, the trend in myotome size was reversed to an extent by 120 hpf (Fig. 4B, right panels). The mutant larval myotomes had a smaller dorsoventral length yet similar overall volume as the wild-type. The number of nuclei was similar, yet there were more cells in the *mymx* mutants. There may be a lower capacity for muscle hypertrophy in the mutants due to a lack of multinucleation, which could result in reduced expansion in size of individual muscle fibres. However, the tissue volume as a whole remained similar due to the increased number of fibres.

### Dynamics of cell rearrangements during myotome formation are perturbed in *mymx*$^{-/-}$ mutants

Do *mymx*$^{-/-}$ mutant myocytes exhibit differences in the dynamics of cell rearrangements in the developing myotome? We performed high-resolution timelapse imaging of somites 16-18 for a period of 12 h, shortly after their segmentation from the pre-somitic mesoderm up to when all myofibres were fully elongated (see Materials and Methods, Fig. 5A). A complete z-stack of the entire myotome was acquired at 3-min intervals, allowing us to follow individual myocytes throughout this entire period. A previously reported drift-correction code was used to rectify for the movement

## A

**Zebrafish *myomixer* coding region:**

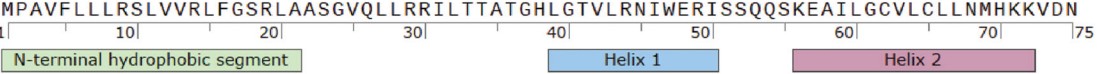

**Myomixer peptide:**

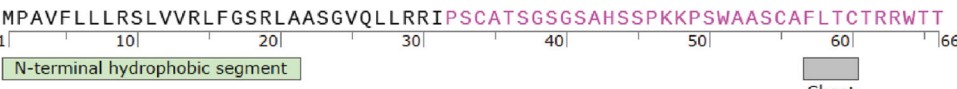

**Myomixer Δ28bp predicted peptide:**

MPAVFLLLRSLVVRLFGSRLAASGVQLLRRI PSCATSGSGSAHSSPKKPSWAASCAFLTCTRRWTT

## B

**Fig. 2. Loss of *mymx* function disrupted embryonic and larval skeletal muscle fibre morphology.** (A) Top: nucleotide sequence of the *mymx* coding region in zebrafish showing the 28 bp deletion (orange) by CRISPR-Cas9 gene editing. Bottom: amino acid sequences and predicted secondary structural elements of native and mutant Mymx peptides, with the altered amino acids of the extracellular domain shown in pink. (B) Wild-type siblings (left column) and *mymx*$^{-/-}$ (right column) embryos at 30, 48 and 120 hpf, immunostained for nuclei (DAPI, green) and cell membranes (β-catenin, magenta) to visualise the skeletal muscle fibres in the developing myotomes. Scale bars: 50 µm. Example cells highlighted with white outlines. Fused cells in *mymx*$^{-/-}$ embryos at 120 hpf highlighted by white arrowheads.

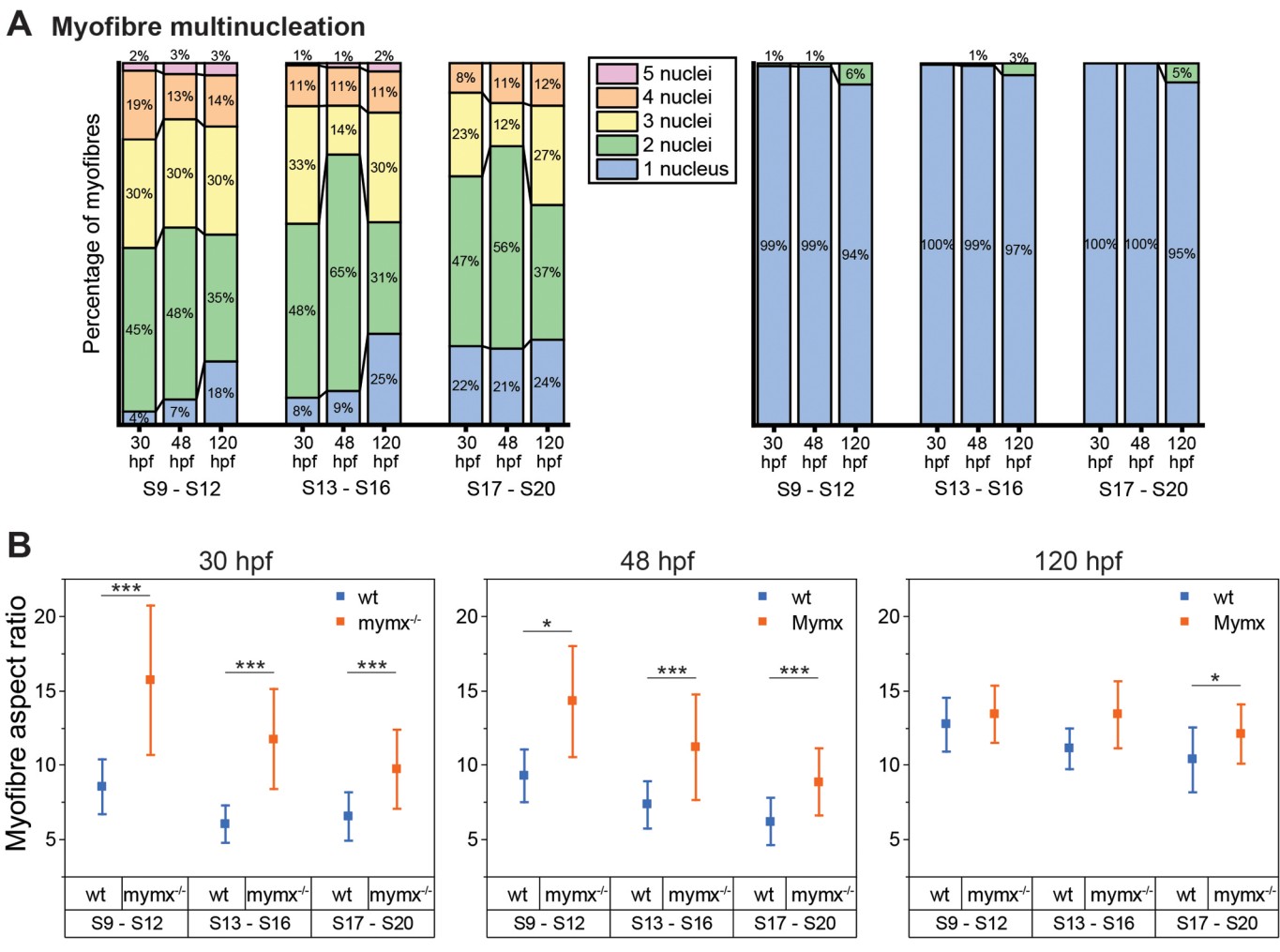

**Fig. 3. Quantification of phenotypic effects on cell structure in *mymx*⁻/⁻ embryos.** (A) Percentage of myofibres with the respective number of nuclei in wild-type (left) and *mymx*⁻/⁻ (right) embryos at 30, 48 and 120 hpf. (B) Aspect ratio (long axis to short axis) of myofibres at different developmental stages in wild-type (blue) and *mymx*⁻/⁻ (orange) embryos. Myotomes grouped by somite numbers (S9 to S12, S13 to S16, and S17 to S20). Error bars represent the standard deviation (*n*=12 myotomes from three embryos). Difference between groups were calculated by two-sided permutation *t*-test and represented as: *$P<0.05$, **$P<0.01$, ***$P<0.001$.

of the tail during embryo growth and align the myotomes over time (Mendieta-Serrano et al., 2022). This allowed us to track the trajectories of myocytes over the duration of the timelapse as they moved around in three-dimensional space.

We have reported that myocytes undergo dynamic rearrangements during myotome formation, but that their speed reduces immediately prior to cell fusion (Mendieta-Serrano et al., 2022). We investigated whether cell rearrangements followed a similar pattern in *mymx*⁻/⁻ embryos. We compared cell speeds between wild-type and *mymx*⁻/⁻ embryos, with cells starting in similar positions (Fig. 5B, left panel). Initially, the cell speeds were in a similar range for mutant and the wild-type. Yet, during the window of 500-700 min after segmentation (Fig. 5B, right panel), the average cell speed was higher in *mymx*⁻/⁻ embryos, perhaps due to a slowdown in cell elongation along the AP-axis (as similarly reported for *mymk*⁻/⁻ embryos).

Thus, the loss of myocyte fusion in *mymx*⁻/⁻ embryos impacted cell dynamics in the developing myotome. Myocytes in *mymx*⁻/⁻ embryos appeared more dynamic, slowing down only once they fully elongated and attached to the AP boundaries of the myotome. Related work has shown that cell dynamics changes in *mymk*⁻/⁻ embryos impacted formation of the myotome segments. Fusion may be playing an

important role in enhancing the rate at which cells span the developing myotome segments (Mendieta-Serrano et al., 2025 preprint).

**Myocyte fusion partially recovered in *mymx*⁻/⁻ mutant adult zebrafish**

It has previously been reported that Mymk is required for muscle fusion throughout the lifetime of the zebrafish, while the adhesion molecules such as Jamb and Jamc that also regulate fusion at the embryonic stage (Powell and Wright, 2011), do not play a role for fusion in the postembryonic and the adult stage (Si et al., 2019; Hromowyk et al., 2020). Is muscle fusion chronically blocked in *mymx*⁻/⁻ mutant adults as in *mymk*⁻/⁻ mutants? To answer this, we studied the morphology of individual muscle fibres isolated from the trunk musculature of 3-month-old fish.

Consistent with *mymk*⁻/⁻ mutants described earlier (Wu et al., 2022), we observed that muscle fibres in *mymx*⁻/⁻ mutant adults were smaller in size than similarly positioned wild-type fibres. However, this size reduction was not as stark as for *mymk*⁻/⁻ adults (Fig. S3A). Quantification of myofibre widths revealed a significant difference between all three genotypes (Fig. S3B). Muscle structure in *mymx*⁻/⁻ fish appeared to have an

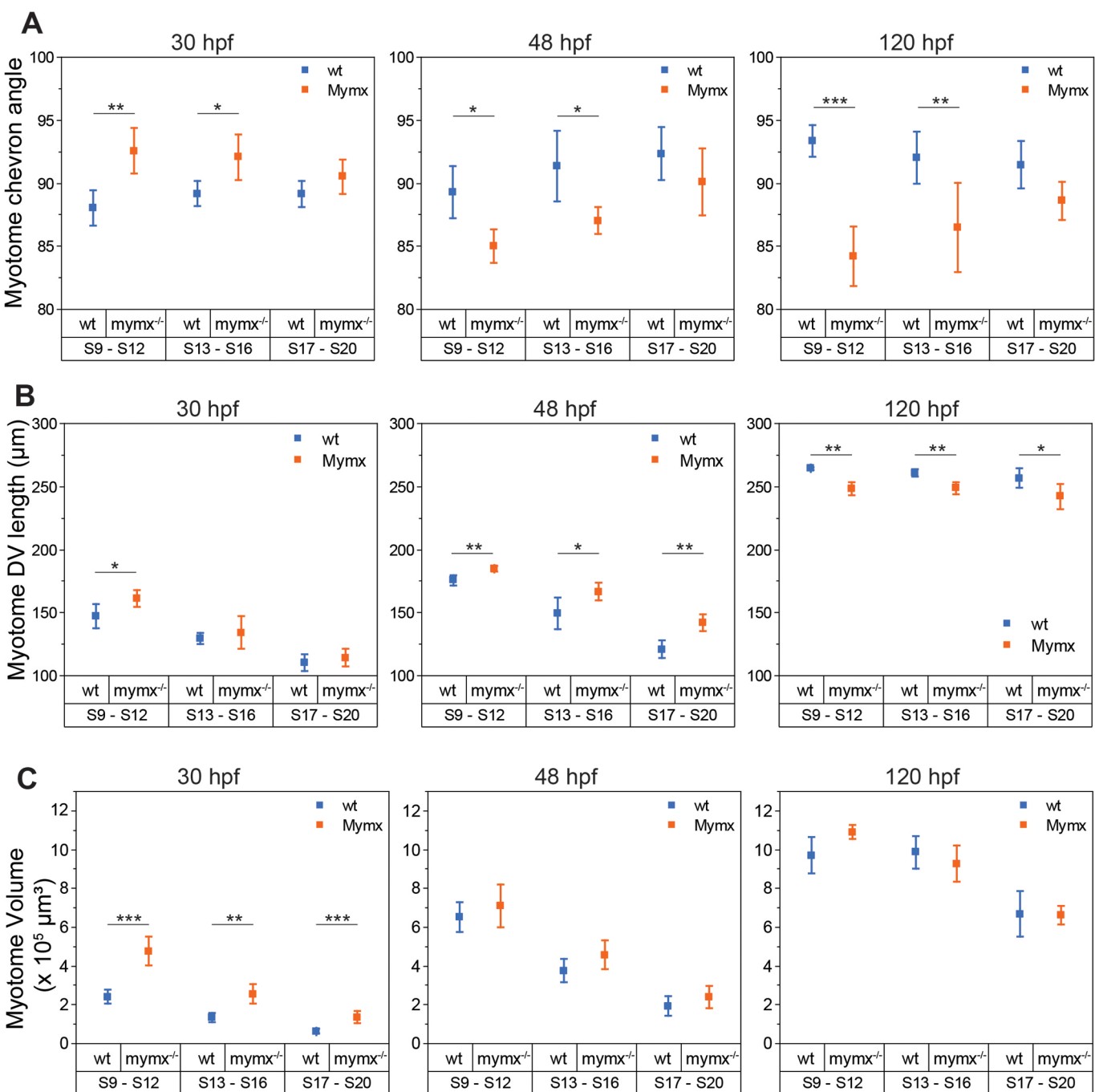

**Fig. 4. Quantification of phenotypic effects on myotome structure in *mymk*<sup>−/−</sup> embryos.** (A) Chevron angle of myotome structure in wild-type (left) and *mymx*<sup>−/−</sup> (right) embryos at 30, 48 and 120 hpf. (B) Myotome length along DV axis, and (C) volume of myotomes in wild-type and *mymx*<sup>−/−</sup> embryos. Myotomes grouped by somite numbers (S9 to S12, S13 to S16, and S17 to S20). Error bars represent the standard deviation (*n*=12 myotomes from three embryos). Difference between groups were calculated by two-sided permutation *t*-test and represented as: *$P<0.05$, **$P<10^{-2}$, ***$P<10^{-3}$.

intermediate phenotype in between the wild-type and *mymk*<sup>−/−</sup> mutants. A similar pattern was also observed in terms of multinucleation of the adult muscle fibres (Fig. S3C). While *mymk*<sup>−/−</sup> myofibres exhibited a very low rate of multinucleation, as reported previously (Si et al., 2019; Hromowyk et al., 2020), *mymx*<sup>−/−</sup> myofibres had a significantly higher number of nuclei, but significantly lower than the wild-type.

Thus, while both Mymk and Mymx are essential for myocyte fusion and proper muscle structure formation in zebrafish embryos, the loss of Mymx is less severe in the larval and adult stages.

## Impact of *mymx*<sup>−/−</sup> on adult zebrafish morphology and skeletal structure

There exists a dynamic interplay between bone and skeletal muscle (Kirk et al., 2020; Herrmann et al., 2020; Sharir et al., 2011; Hamrick et al., 2010), which arises through multiple mechanisms including the mechanical forces generated by muscle contractions during development and in adult life (Herrmann et al., 2020; Nowlan et al., 2010). The power of these muscle contractions can influence the formation and remodelling of bone as an adaptive response (Sharir et al., 2011). Human patients with muscular

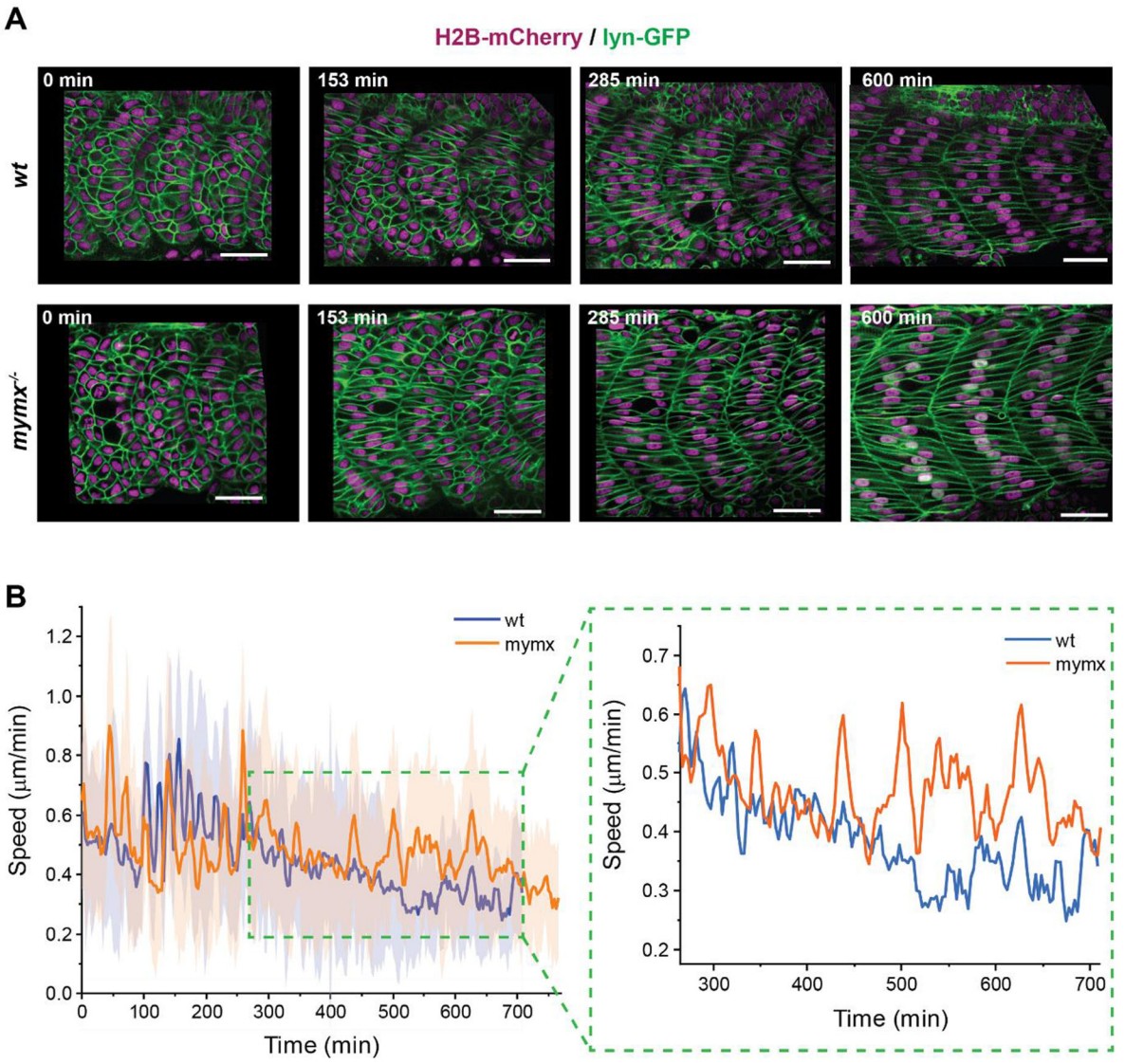

**Fig. 5. Dynamics of muscle precursor cell rearrangements during early myogenesis.** (A) Timelapse imaging of myotome development in a wild-type (top) and a *mymx*⁻/⁻ (bottom) embryo injected with H2B-mCherry and lyn-GFP mRNAs to visualise nuclei (magenta) and cell membranes (green), respectively (scale bars: 30 µm). (B) Average cell speed from wild-type and mutant embryos, starting out in similar positions in the myotome (*n*=40 cells from four myotomes). The solid line represents the mean and shaded areas represent the standard deviation. Right panel shows an inset of the window within the green box, marking the period during which most of the initial fusion events occurred in wild-type embryos. Error bars omitted for clarity (shown on left). The speed difference is significant (*P*<0.01) between 500-700 min (two-sided *t*-test, grouping the data into 100 min bins, with four independent samples).

dystrophies or osteoporosis often display both reduced muscle and skeletal integrity (Ferrucci et al., 2014)*.* Zebrafish has been used as a model for skeletal diseases (Le Pabic et al., 2022), such as osteoporosis (Bergen et al., 2019; Masiero et al., 2024). While there is a lack of quantitative study of the interplay between myocyte fusion and bone formation (Bosco et al., 2021; Herrmann et al., 2020), the zebrafish provides an excellent system for studying this further (Chatani et al., 2011; Dalle Carbonare et al., 2025).

We hypothesised that defects in muscle fibre differentiation due to loss of myocyte fusion will result in disrupted adult skeletal structures. We first compared the overall morphology of adult *mymk* and *mymx* mutants with wild-type zebrafish (Fig. 6A, Fig. S4). As described previously, we found that homozygous *mymk*⁻/⁻ fish exhibited a markedly emaciated, small and scoliotic morphology while the *mymx*⁻/⁻ mutants were more similar to the wild-type (Fig. 6B, Fig. S4). To test the possibility that impairment in myocyte

fusion can impact skeletal morphology, we generated whole-mount Alizarin Red preparations of 3-month-old adult wild-type, *mymk*⁻/⁻ and *mymx*⁻/⁻ zebrafish skeletons (Fig. 6). Formation of craniofacial features can be sensitive to external factors (Raterman et al., 2020). We compared the skull and cranium shape between wild-type and the fusion mutants, and this revealed obvious craniofacial abnormalities in the *mymk*⁻/⁻ mutant fish (Fig. 6C). This effect mirrors human Carey-Fineman-Ziter Syndrome, with affected patients displaying similar craniofacial abnormalities likely from muscle weakness due to mutations in *MYMK* (Hedberg-Oldfors et al., 2018; Liang et al., 2024; Di Gioia et al., 2017). Notably, the *mymk*⁻/⁻ adults exhibited a rounded cranium, reduced premaxilla, and enlarged mandibles in comparison to wild-type animals. Interestingly, we did not detect large craniofacial defects in the *mymx*⁻/⁻ adults; see below for more detailed analysis. This result is somewhat surprising, given that Mymx has recently been shown to

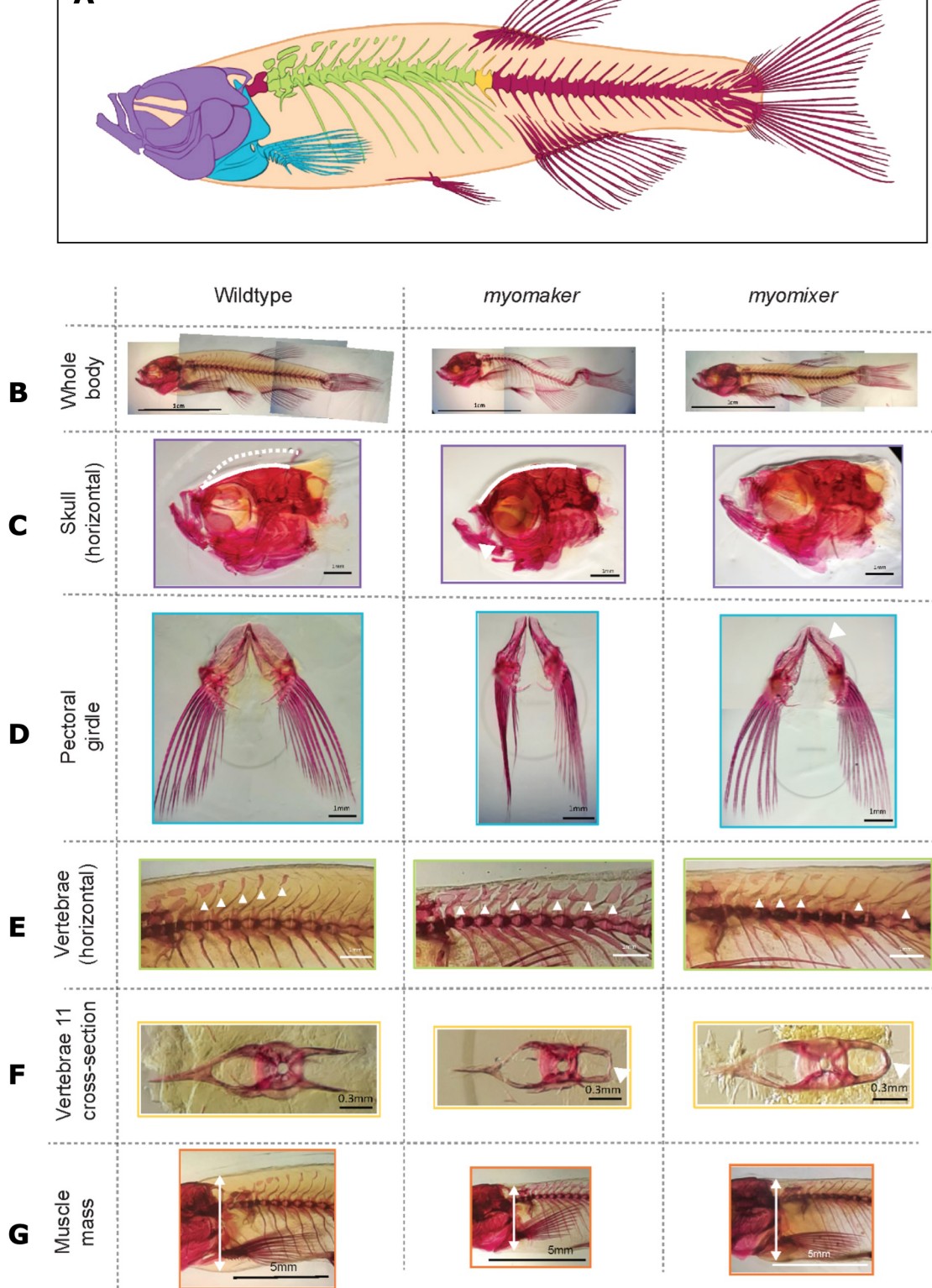

**Fig. 6.** See next page for legend.

be necessary for myocyte fusion during craniofacial muscle formation in the zebrafish embryo (Zhang et al., 2025).

We also observed abnormalities in the structure, size, shape and porosity of the coracoid and cleithrum bones, which comprise the

pectoral girdle (Grandel and Schulte-Merker, 1998), in both fusion mutants (Fig. 6D). These bones, which typically lie below the operculum, were narrower and more porous than those seen in the wild-type. Both mutants displayed reduced surface area, and minor

**Fig. 6. Skeletal structure of wild-type and fusion mutant adult zebrafish.**
(A) Diagram of the adult zebrafish skeleton in lateral view. Coloured elements correspond to the position of skeletal elements depicted in C (skull, purple), D (pectoral girdle, blue), E (precaudal vertebrae, green), F (1st caudal vertebrae cross-section view, yellow) and G (muscle mass comparison, orange). (B) Whole-body comparisons of 3-month-old wild-type, $mymk^{-/-}$ and $mymx^{-/-}$ fish cleared and stained with Alizarin Red. (C) Comparison images of the horizontal view of the skull. Solid white lines trace the shape of the cranium, beginning behind the premaxilla. Dotted line represents the traced shape of the cranium in the $mymk^{-/-}$ fish compared to wild-type. White arrowhead points to abnormal mandibular structure in $mymk^{-/-}$ fish. (D) Comparison of the pectoral girdle size and structure. The white arrowhead indicates left-right asymmetry in the pectoral girdle of $mymx^{-/-}$ mutants. (E) Horizontal view of precaudal vertebrae and supraneural bones compared across genotypes. White rings indicate overgrowth of neural spines of precaudal vertebrae and increased supraneural/ectopic bone growth in $mymk^{-/-}$ and $mymx^{-/-}$ fish. (F) Comparison images of the 1st caudal vertebra. White arrows indicate fusion. (G) Muscle mass size of 3-month-old zebrafish. White arrows indicate the overall bulk of muscle behind the skull. $n$=3 fish for each genotype.

left-right asymmetry of the structure was also notable in $mymx^{-/-}$ adults.

Comparisons of precaudal vertebrae and supraneural bones across genotypes highlighted abnormal ectopic bone growth in $mymk^{-/-}$ and $mymx^{-/-}$ adults (Fig. 6E,F). Abnormalities in neural arch structure and in supraneural number, size and surface area were particularly severe in $mymk^{-/-}$ adults. We observed similar, though less severe, abnormalities in neural arch structure in $mymx^{-/-}$ adults. The severity of the phenotypes correlated with reduced muscle mass, which may lead to reduced power in generating locomotion, leading to ectopic bone growth as a possible compensatory mechanism. We also observed unusual fusion of neural spine projections in precaudal vertebrae in both fusion mutants that were not present in the wild-type (Fig. 6F). We further compared muscle mass behind the skulls (Fig. 6G). We found severely reduced muscle mass in $mymk^{-/-}$ adults. Consistent with our results above on myofibre morphology, $mymx^{-/-}$ adults displayed an intermediate phenotype, with muscle mass thickness between wild-type and $mymk^{-/-}$ adults.

To analyse these observations in more depth, we focused on two interesting points: (1) the formation of ectopic bone growth in the back; and (2) changes in the craniofacial morphology (Fig. 7A). We found that the ratio of bone mass in the vertebrae increased significantly from wild-type in both $mymk$ and $mymx$ mutant embryos (Fig. 7B). The skull was extended moderately but reproducibly in the $mymx$ loss-of-function fish (Fig. 7C). For $mymk$ deficient fish, we observed a very large variation in the measures, consistent with the severe phenotype observed in these fish. Measuring the curvature of the skull, we did not notice any significant difference between genotypes, though this could be due to the low $n$ (three per condition).

## DISCUSSION
Multinucleation is thought to enable an amplified degree of transcriptional activity that is required for building and maintaining the highly structured contractile apparatus of muscle fibres required for force generation. Myocyte fusion is not only essential for the generation of multinucleated muscle fibres during the embryonic episode of myogenesis but also contributes to hypertrophic growth of muscle tissue during juvenile stages and in adult life. In addition, fusion also plays a critical role in regenerative restoration of muscles in response to injury. These post-embryonic phases of myogenesis involve the fusion of myocytes generated

from skeletal muscle stem cells, the satellite cells, with preexisting muscle fibres or the *de novo* generation of new myotubes through the fusion of myocytes among themselves.

The fusogenic proteins, Mymk and Mymx, are instrumental in driving myocyte fusion in all vertebrate species examined, and in humans, their dysfunction significantly compromises skeletal muscle development as exemplified by the pathological manifestation in individuals afflicted with Carey-Finneman-Ziter syndrome. Interestingly, the fusogenic ability of these proteins can also be harnessed to deliver therapeutic payloads to diseased muscle tissue (Hindi et al., 2023). Given these clinical ramifications of Mymk and Mymx function, it is imperative that we further our understanding of the biology of these fusogens. We have now quantitatively elucidated the spatiotemporal expression pattern of the *mymx* gene in the developing zebrafish myotome and examined the function of the Mymx protein at different stages of myogenesis in this organism. By resolving *mymx* transcript distribution at a higher resolution than previously reported, we could identify a temporal wave of expression from the medial to the lateral aspect of the developing myotome, limited to specific layers of fast myocytes at any given time. This pattern correlated with the spatiotemporal pattern of cell fusion events that we have described earlier (Mendieta-Serrano et al., 2022). It has been implied that the levels of *mymk* expression in the myotome is affected by Shh signals from the notochord (Ganassi et al., 2018), but we have previously observed that the mediolateral pattern of expression is not significantly altered in mutants with defects in Shh signalling (Mendieta-Serrano et al., 2022). Further investigation will be needed to understand whether *mymx* expression pattern is intrinsically defined by the differentiation state of the myocyte, as it seems with *mymk*, or influenced by environmental signals, such as morphogens. It remains unclear whether loss of fusion impacts cell fate, in particular whether there is an impact on the proportions of fast- and slow-twitch fibres. In the fusion-deficient mutants, the fast fibres superficially resemble slow fibres. It will be interesting to explore further whether these fibres are functionally similar to slow muscles, e.g. imaging the domains of Prdm1a expression. Relatedly, future work can follow expression of *mymx* throughout development, including larval stages, to build a more complete picture of fusion events post the initial wave during embryogenesis.

We report a detailed characterisation of the impact of a *mymx* loss-of-function mutation on muscle formation at different stages of zebrafish development. The loss of Mymx abrogated myocyte fusion at the embryonic stage, which extended over the period of dynamic cell movements within the myotome, compared to wild-type embryos. Mymx loss also significantly impacted the structure of muscle fibres and the myotome tissue as a whole during the embryonic as well as the larval stages. It has been reported that active stresses generated by the rearrangements and elongation of myocytes and muscle fibres are responsible for forming a robust myotome structure (Tlili et al., 2019). Therefore, the absence of myocyte fusion in $mymx^{-/-}$ mutants could be envisaged to alter these cell-generated forces, impacting the development and morphology of the tissue structure as a whole.

We found an interesting behaviour in terms of muscle segment growth between embryonic and later developmental stages of *mymx* embryos. We observed hypertrophy during embryonic stages when Mymx is expressed (Fig. 3). This may be due to some (unknown) "compensatory mechanism". In contrast, at 120 hpf when Mymx expression has substantially declined, there were changes in myotome shape, with reduced DV extent and chevron angle

**Fig. 7. Quantification of skeletal structure changes between wild-type and fusion mutant adult zebrafish.** Definition of measures used for defining the bone to mass ratio (A) and the skull length and curvature (A′). (B) Comparison of bone to mass in the three genotypes (*n*=3). (C) Comparison of skull length (*n*=3). Two-sided *t*-test performed for *P*-values and graphs generated using estimationstats.com (Ho et al., 2019).

(Fig. 4). The increased number of muscle fibre attachments at the myotome boundaries in *mymx* mutants (due to lack of fusion) could potentially generate increased contractility, reducing the somite extent. Further, there is resumption of fusion during larval stages

(though less than in the wild-type). It will be interesting to explore the mechanism(s) underlying these observations in future work.

Mymk is critical for muscle fusion and regeneration throughout the lifetime of the zebrafish (Shi et al., 2018). On the other hand, the

Biology Open

cell adhesion molecules, Jamb and Jamc, are required for fast myocyte fusion at the embryonic stage, but are completely dispensable at the larval and adult stages (Si et al., 2019). Mymk is essential for fusion and proper tissue morphogenesis at the embryonic stage, and its loss is only partially rescued in the larval and adult stages of muscle growth. In agreement with previous reports, we find that the severity of our $mymx^{-/-}$ mutant lies in between the $mymk^{-/-}$ phenotype and the wild-type. However, in mammals, both Mymk and Mymx are critical for myocyte fusion during embryonic development as well as at the adult stage (Millay et al., 2013, 2014; Bi et al., 2017, 2018; Quinn et al., 2017; Zhang et al., 2017). These observations, together with analysis of the evolutionary history of these fusogen genes, support the notion that Mymk arose in the chordate lineage as an essential factor for myocyte fusion while the emergence of Mymx occurred later, in the early vertebrates, perhaps to increase the efficiency of fusion process and became indispensable in the mammals (Zhang et al., 2022).

Finally, we have taken advantage of the unique ability of the zebrafish $mymk$ and $mymx$ mutants to survive into adulthood to explore the effect of impaired myogenesis on the development of the vertebrate skeleton. Our analysis with the different myocyte fusion perturbation conditions revealed interesting differences. While both $mymk^{-/-}$ and $mymx^{-/-}$ fish displayed deformed, porous pectoral girdles and excess bone deposited at pre-caudal neural spines, the craniofacial features of $mymx^{-/-}$ fish were largely unaffected, underscoring a correlation between the severity of myocyte fusion defect with deformities of the bone. It will be interesting to perform a longitudinal study of the developing skeleton to more precisely map when and where these skeletal defects emerge when myocyte fusion is disrupted, including the use of live markers to follow osteoblasts. Furthermore, reduced muscle mass is typically associated with reduced bone formation and integrity. Yet surprisingly, we see here that our fusion deficient zebrafish featured overproduction of bone. Is this in part to provide structural integrity to compensate for reduced muscle mass and power? Alternatively, could it imply that the myocyte fusion proteins are also involved in the fusion of monocyte/macrophages into multinucleated osteoclasts that function in bone remodelling?

Since our pioneering study that established the zebrafish as a facile model for studying vertebrate myocyte fusion (Roy et al., 2001), a number of investigators have exploited this system to investigate the cellular and genetic basis of the fusion process (Powell and Wright, 2011; Luo et al., 2022; Srinivas et al., 2007; Si et al., 2019). Importantly, we and others have shown that the myocyte fusogens, Mymk and Mymx, first identified in the mouse, also play a conserved role in driving myocyte fusion in the zebrafish (Zhang and Roy, 2017; Landemaine et al., 2014; Wu et al., 2022). Mutants with loss of $mymx$ still maintain $mymk$ expression, raising the possibility that the activity of Mymk in the $mymx$ mutants may compensate partially to enable low rates of fusion (Yong et al., 2024). Our recent $in\ toto$ analysis of myocyte fusion in the living zebrafish embryo has enabled imaging of vertebrate myocyte fusion dynamics in real time that is not currently achievable with the mouse (Mendieta-Serrano et al., 2022, 2025). Our present analysis of zebrafish $mymx$ expression and function at cellular resolution should enable further investigations into the basic biology of myocyte fusion in the early embryo. This works also provides insight into how loss of fusion impacts skeletal form of the adult fish and has relevance for understanding the pathological mechanisms underlying myopathies that arise from aberrations in myocyte fusion.

## MATERIALS AND METHODS

### Zebrafish husbandry
All zebrafish strains were maintained according to standard procedures for fish husbandry at the zebrafish facilities of the Institute of Molecular and Cell Biology and Warwick University. All experiments with the zebrafish in Singapore were approved by the Singapore National Advisory Committee on Laboratory Animal Research (protocol number 221702). All experiments in Warwick were performed in compliance with the University of Warwick animal welfare and ethical review board (AWERB) and the UK home office animal welfare regulations, covered by the UK Home office licenses PEL 30/2308 and X59628BFC to the University of Warwick. We utilised the previously established strains $Tg(mylpfa:H2B-eGFP)$ and $Tg(mylpfa:H2B-eGFP);mymk^{sq36}$ in this study, alongside the newly generated $mymx^{-/-}$ loss-of-function mutant.

### Fluorescence *in situ* hybridisation
Fluorescent *in situ* hybridization of whole-mount zebrafish embryos was performed as previously described (Narayanan and Oates, 2019). The coding region of $mymk$ and $mymx$ were cloned from zebrafish cDNA by PCR amplification using the primers:
mymk-forward: ATGCGGATCCCGCAATGGGAGCGTTTATCGCCAAG
mymk-reverse: ATGCGAGCTCTACACAGCAGCAGAGGGTGTAG
mymx-forward: ATGCGAATTCCAAAATGCCAGCCGTTTTCCT-CTTGC
mymx-reverse: ATGCGAGCTCAGTCTTGTTGTCTCGCGTGTAATT.
Digoxigenin-labelled RNA probes were transcribed using the DIG RNA Labeling Mix (Roche, 11277073910). Zebrafish embryos were fixed at the 22-somite-stage using 4% paraformaldehyde in 1X PBS for 2 h at room temperature. Embryos were then dehydrated in methanol, permeabilised using proteinase-K treatment, and hybridised using the DIG-labelled RNA probes at 60°C for 16 h. Fluorescent staining was performed using an anti-digoxigenin-AP antibody (Roche, 11093274910) and SIGMAFAST Fast Red (Sigma, F4648). The nuclei were labelled using DAPI (Sigma, D9542).

### Generation of a *mymx* knockout mutant
A stable $mymx$ knockout mutant line ($mymx^{-/-}$) was generated using CRISPR-Cas9. sgRNA targeting a 20 bp region in exon 1 of the zebrafish $mymx$ gene was synthesised using PCR amplification. A mixture of 1000 ng/µl of the sgRNA and 800 ng/µl of Cas9 protein (ToolGen) was incubated at 37°C for 15 min. 0.5-0.75 nl of the mixture was injected into Tg(mylpfa:H2B-eGFP) embryos at the 1-cell stage. Injected embryos (F0 generation) were raised to adulthood and fin-clipped at 3 months of age for genotyping. The following primers were used for PCR amplification of the $mymx$ gene:
Forward: ATGCCAGCCGTTTTCCTCTTGC
Reverse: AGTCTTGTTGTCTCGCGTGTAATT.
Amplification products were cloned into pCRII-TOPO vectors and sequenced to identify mutant alleles of the $mymx$ gene. Fish with a 28 bp deletion allele at the target region (Fig. 2A) were identified and outcrossed with Tg($mylpfa:H2B-eGFP$) fish (Zhang and Roy, 2016). These embryos (F1 generation) were raised to adulthood and genotyped at 3 months of age to identify those carrying the 28 bp deletion allele. Identified heterozygous pairs were in-crossed and the embryos (F2 generation) were raised and genotyped to confirm the presence of the mutant allele. The identified F2 generation fish were used to establish a stable line of $mymx^{-/-}$ mutants.

### Immunostaining of whole embryos
Embryos were dechorionated using forceps and fixed with 4% paraformaldehyde in 1X PBS for 2 h at room temperature. The samples were then dehydrated in methanol at −20°C and blocked using 5% BSA in 1X PBS for 4 h. They were immunostained using an anti-β-catenin primary antibody (Abcam, ab6301) and an anti-mouse IgG AlexaFluor 488 secondary antibody (Invitrogen, A28175), and co-stained with DAPI (Sigma, D9542).

### Dissection of and immunostaining of adult muscle fibres
Dissection and immunostaining of adult muscle fibres was performed using a modified version of previously reported protocol (Shi et al., 2018). 3-

Biology Open

month-old fish were euthanised using an overdose of Tricaine and fixed using 4% paraformaldehyde in 1X PBS for 1 h at room temperature. The skin was carefully removed using forceps and fresh 4% paraformaldehyde was added for 30 min. After washing with 1X PBS, whole muscle bundles were dissected using forceps and placed in 0.04% saponin in PBS in a glass dish. Individual muscle fibres were carefully detached and cleaned up using PBDT (1X PBS with 1% BSA, 1% DMSO and 0.5% Triton X-100), before being placed in blocking buffer (5% BSA in PBDT) and kept for 16 h at 4°C with gentle rocking. Muscle fibres were immunostained using an anti-α-catenin primary antibody (Sigma, A7732), an anti-mouse IgG AlexaFluor 488 secondary antibody (Invitrogen, A28175), and co-stained with DAPI (Sigma, D9542). Stained muscle fibres were then mounted in glycerol on glass slides with #1.5 thickness coverslips.

## Alizarin Red staining
Skeletal stains of adults were performed according to Potthoff et al. (1984), with minor modifications to the protocol detailed in (Westerfield, 2000). 3-month-old adult fish were euthanised in Tricaine solution and were subsequently fixed in 4% paraformaldehyde in 1X PBS for 24 h. Specimens were then dehydrated for 24 h in 100% ethanol, soaked in saturated sodium borate solution overnight, and then bleached in 0.5% H2O2/0.5% KOH solution for 1.5 h. Muscle tissue was cleared by trypsin digestion (1% trypsin in 35% saturated NaBO4) for 24 h, until the body achieved 60% clarity. Bones were stained by incubation in a 0.05% Alizarin Red/0.5% KOH solution for 24 h. Trypsin digestion was then repeated to de-stain and further clear the fish, followed by incubation in a graded glycerol series (0.5% KOH/30% glycerol, 0.5%KOH/60% glycerol) for 12 h in each solution to further clear the samples. Fish were then de-scaled by gentle scraping with forceps under a microscope and transferred to 100% glycerol for long-term storage at room temperature. All incubations occurred at room-temperature with gentle rocking unless otherwise described.

## Confocal microscopy
The immunostained samples were equilibrated in glycerol and deyolked using dissection needles. They were mounted onto glass slides with #1.5 thickness coverslips. Samples were imaged on a Nikon A1R confocal microscopy with a 40X water immersion objective. Z-stacks of the entire myotomes were acquired at 1 μm steps.

Microinjected embryos at the 20-somites-stage were dechorionated using forceps and maintained in embryo medium containing 200 μg/ml tricaine. Embryos were embedded in 1% low-melting-point agarose on a glass-bottom Petri dish. The solidified agarose surrounding the trunk of the embryo was carefully removed using a needle to allow the trunk to elongate normally and grow unrestricted for the 12-h imaging period.

Timelapse imaging was performed on a Nikon Eclipse Ti microscope with a Yokogawa CSU-W1 confocal spinning disk unit, with a 60X water immersion objective. A z-stack of the entire myotome was acquired at 1 μm steps, at an interval of 3 min for a period of 12 h. To correct for movement during tail growth, somite alignment was performed on Fiji using a custom macro (Mendieta-Serrano et al., 2022).

## Image analysis and statistics
Quantifications of fluorescence intensities, bone area, and cell and myotome morphologies were performed using Fiji (Schindelin et al., 2012). Cell trajectories were tracked using the Fiji plugin MTrackJ (Meijering et al., 2012) by manually annotating the centroid of the nucleus of each cell at each time point of the timelapse movies. Cranium curvature analysis was performed using the Fiji Kappa tool (Mary and Brouhard, 2019 preprint).

Statistical significance of the difference between groups was determined by two-sided permutation *t*-test. Sample sizes for each experiment are mentioned in the figure captions.

## Acknowledgements
We thank the zebrafish and central imaging facilities at the IMCB and Warwick University for zebrafish husbandry and imaging resources, respectively, and D. Millay for helpful discussions. S.T. was recipient of an A*STAR-Warwick University research attachment program (ARAP) fellowship.

## Competing interests
The authors declare no competing or financial interests.

## Author contributions
Conceptualization: S.D., S.T., T.E.S., S.R.; Data curation: S.D., S.T.; Formal analysis: S.D., S.T., T.E.S.; Funding acquisition: T.E.S., S.R.; Investigation: S.D., S.T., H.L.Y., S.R.; Methodology: S.T., H.L.Y.; Project administration: T.E.S., S.R.; Resources: H.L.Y.; Supervision: T.E.S., S.R.; Visualization: S.D., S.T.; Writing – original draft: S.D., S.T., T.E.S., S.R.; Writing – review & editing: T.E.S., S.R.

## Funding
This work was supported by funds from the Biotechnology and Biological Sciences Research Council (BBSRC) Responsive Mode Grant (BB/W006944/1) to T.E.S. and the Institute of Molecular and Cell Biology (IMCB), A*STAR to S.R. Open Access funding provided by University of Warwick. Deposited in PMC for immediate release.

## Data and resource availability
All relevant data and details of resources can be found within the article and its supplementary information.

## Peer review history
The peer review history is available online at https://journals.biologists.com/bio/lookup/doi/10.1242/bio.062305.reviewer-comments.pdf

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
