## [Peer Review File · Biology Open]

Dynamic expression and differential requirement of the myocyte fusogen Myomixer during distinct myogenic episodes in the zebrafish

Sunandan Dhar, Serena Thomas, Hui Li Yeo, Sudipto Roy and Timothy Saunders
DOI: 10.1242/bio.062305

Editor: Tristan Rodríguez

Review timeline

Submission to sister journal:	1 September 2025
Editorial decision at sister journal:	2 October 2025
Transfer to Biology Open:	17 November 2025
Accepted:	18 November 2025

Original submission to sister journal

First decision letter

MS Title: Dynamic expression and differential requirement of the myocyte fusogen Myomixer during distinct myogenic episodes in the zebrafish

Authors: Sunandan Dhar; Serena Thomas; Hui Li Yeo; Timothy Saunders; Sudipto Roy
Article Type: Research Article

Dear Timothy,

I have now received all the referees' reports on the above manuscript, and have reached a decision. I am sorry to say that the outcome is not a positive one. The referees' comments are appended below, or you can access them online: please go to

As you will see, the referees consider that there are insufficient conceptually new findings in your study to justify publication in this journal. Given these opinions, I must therefore, reject your paper.

I do realise this is disappointing news, but we receive many more papers than we can publish, and we can only accept manuscripts that receive strong support from referees. I do hope you find the comments of the referees helpful, and that this decision will not dissuade you from considering our journal for publication of your future work. Many thanks for sending your manuscript to us.

Reviewer 1

Summary and Potential Significance:

The manuscript of Dahr et al. mainly describes roles of the myocyte fusogen gene myomixer (mymx) in skeletal muscle development in zebrafish. Further data on myomaker (mymk), another fusogen gene, is presented and extends the study from skeletal muscle to skeleton biology. The

work relies on a combination of genetics, in situ hybridization, immunohistochemistry, established transgenic reporter lines, imaging, and statistics.

It was previously shown that myocyte fusion as an essential process during the establishment of the myotome is regulated by *mymx* and *mymk*. Compared to these previous studies, the authors reveal more detail about the spatiotemporal expression patterns of *mymx* at 24 hours post fertilization and assay many aspects of fibre biology at 30, 48 and 120 hpf, comparing wild type to a newly generated *mymx* mutant. In the mutant fast fibres seemed to suffer from a fusion deficit as they remained mononucleated. The authors show that in adult fish reduced nuclei number and fibre dimension are not only reduced in the *mymx* but also in a previously generated *mymk* mutant. Lastly, the authors investigate skeletal morphogenesis in both mutants. In the abstract the authors state they found that "impaired muscle differentiation due to defects in myocyte fusion caused significant defects in skeletal morphogenesis". However, it remains unclear whether the described skeletal defects observed in the mutants are a direct consequence of defects in the myotome, unless the authors perform conditional knock-outs in the myotome without an impact for the skeleton.

Although the data is beautifully presented, the study seems to be largely redundant with the previously published study by Shi et al., 2017 and bears the additional caveat mentioned above.

SUGGESTIONS TO AUTHORS

Comments for the Authors:

Although fibre counts are lacking for the *mymx* mutant, the observed mononucleated fibres indeed suggest a fusion deficit. Fast fibres resemble slow fibre in many aspects (single, centralized myonuclei in fibres, fibre orientation) and it would be interesting to know whether these fibres also adopt a slow fate identity on the molecular level.

Fig. 2: It remains untested in this study whether the mutant *Mymx* protein is lost, thus we recommend to change the title of Fig. 2 to "Loss of *mymx* function...". Please consider this caveat throughout the manuscript.

Fig. 3/4: These figures seem to represent the quantifications of data depicted in Fig. 2. It might make sense to pick the most telling results and combine these into one panel, while other results go into the supplementary material. Please insert "wt" and "*mymx*-/-" into panel A of Fig. 3.

Please list the previously established *mymk* mutant and transgenic lines (H2B-mCherry, lyn-GFP) with adequate allele designation in the methods section under "Zebrafish husbandry".

Please follow the zebrafish nomenclature convention throughout text and figures (gene=*mymx*, protein=*Mymx*). See here: <https://zfin.atlassian.net/wiki/spaces/general/pages/1818394635/ZFIN+Zebrafish+Nomenclature+Conventions>

Reviewer 2

I am puzzled by the author's claimed "gap" in knowledge, that the expression and function of *Mymx* are not well understood. The authors state this supposed gap both in the abstract (lines 20-22) and in the introduction (lines 92-93). To substantiate this claim, they will need more clarity on what remains unknown about *mymx*.

The requirement for *mymx* in muscle cell fusion was known in zebrafish at least as far back as 2017, with a starting paper from Eric Olsen's lab with further work coming from Elizabeth Chen and Shaojun Du's groups. The current author's subsequent claims of novelty also run into trouble. Wu 2022 showed that *mymx* is required for growth to normal adult sizes and for myofiber growth too. The only part of the authors conclusions that seems potentially new is the impact of *mymx* on skeleton, which is not explored quantitatively in the current work.

Major comments:

-The authors provide a brief exploration of *mymx* gene expression patterns using fluorescent ISH instead of the previously published DIG ISH. This is not a comprehensive examination and only explores an N=3 animals at a single stage of development (24 hpf). The authors did assess expression across a multitude of ROI at this stage, but that does not change the fact that there are three animals, one stage, and not many believable surprises.

-They provide abundant quantification of the number of myonuclei found in different somite levels in *mymx* through the first week of life. (The WT fusion levels have already been shown in other studies). This reveals that muscle cells occasionally fuse in *mymx*^{-/-}. Rare fusion events like these in *mymx*^{-/-} were previously demonstrated in Wu et al., 2022 "Loss of Myomixer Results in Defective Myoblast Fusion, Impaired Muscle Growth, and Severe Myopathy in Zebrafish"

-The authors show that *mymx* mutants have somites that are briefly taller than WT, but they do not make clear why this transient finding matters.

-The authors provide stills from a time-lapse of cell fusion in the *mymx* mutant. In these stills one can see that the mutant does not fuse, which already well known. Furthermore, a higher-resolution time-lapse of muscle cell fusion in another *mymx* mutant was provided previously by Luo et al., 2022. In an attempt to make new insights, the authors provide new quantification (panel B) in which the error bars between WT and *mymx*^{-/-} overlap at every timepoint, they then zoom in on a region where the two time-lapse videos might differ and remove the error bars. The lack of error bars in this inset borders on deceptive.

-The authors provide examples of isolated myofibers from adult *mymx* mutants (presumably fast-twitch ones). They then show that *mymx*^{-/-} mutants have a strong, but partial fusion defect that is less severe than *mymk*^{-/-}. However, this finding has also been shown previously with high similarity to Wu 2022, Figure 9.

- Finally, in Figure 7, the authors make an arguably new claim. They provide a non-quantitative exploration of adult skeletal defects in the fusion deficient *mymk*^{-/-} and *mymx*^{-/-} mutants, with no mechanistic insights or tests.

Punchline: To the best of my knowledge, Development does not publish confirmatory studies. I have difficulty finding suggestions to that authors on how to amend this work to move into new turf, since the current story is so directly in line with known materials.

Reviewer 3

In this manuscript, Dhar and colleagues set out to study *Mymx* expression pattern in early developing zebrafish embryo and subsequently generated a *Mymx* knockout line. Using this mutant, they have characterized the developmental cellular and tissue-level defects, such as myofiber morphology, myonuclei number in myofiber and cell rearrangement dynamics. Using single-fiber isolation, they showed adult *Mymk* and *Mymx* mutant both have a reduction of nuclei number per fiber and delineated their severity. Moreover, authors showed the skeletal defects of adult *Mymk* KO zebrafish, but they found that *Mymx* KO has a very mild phenotype. Based on these findings, the authors concluded that myocyte fusion plays an important role in tissue morphogenesis and thus acquisition of the final adult form.

SUGGESTIONS TO AUTHORS

Overall, the manuscript is descriptive and lacks new mechanistic insights on how loss of *Mymx* causes secondary cellular defects in addition to perturbed fusion. Some phenotypic analysis was detailedly executed such as myotome angle, myotome volume and adult myonuclei number which has shed light on the essential role of *Mymx* during myogenesis. However, the rationale of analyzing the cell rearrangement dynamics in *Mymx* mutant is not clear. Also, *Mymx* KO mutant in this manuscript has a largely overlapping phenotype with the previously reported mutants, thus overall there is unclear new knowledge that would be valuable for the field.

Below are specific comments about the current data presented, but without a better narrative with more obvious advances for the field, I do not think this manuscript is optimal for Development.

1. The abstract says that Mymx function is less well understood. I think it is incorrect since there are papers specifically deciphering Mymx activity but these are lacking for Mymk.
2. Line 138: "mono-nucleated slow-twitch muscles that do not express mymk or mymx" may be inaccurate. It was reported Mymk transcripts are detected transiently in slow muscle precursors adaxial cells (Yong, 2024).
3. In Fig. 1, Mymx expression along AP axis is only shown for the 24-hpf but Fig. 1D shows the fluorescent intensity across 5 timepoints. As reported in Shi. et. al, 2017, 24-hpf is not the timepoint with highest Mymx expression. Should authors show all Mymx staining along AP axis to complement the signal intensity quantification? Following that, can the authors describe the Mymx expression pattern and highlight the relevance?
4. Line 148, "mymx has a complicated pattern of expression" is ambiguous. It would be great if the authors could explain the potential relevance, if any, of this expression pattern.
5. Line 200, "low level of myocyte fusion can occur..." based on the presence of binucleated cells. Are binucleated cells definitive or could this be cells that are undergoing cell division?
6. Since the authors have already shown nuclei number per myofiber, it is unclear what the analysis of number of nuclei per myotome segment (Figure 4E) is achieving.
7. Line 249-252, it is unclear how a smaller dorsoventral length and similar overall volume represent a lower capacity for muscle hypertrophy and reduced expansion in size of the muscle fibers. It is confusing because the authors say that overall volume is similar but there is reduced expansion in size. Overall, the concepts to be conveyed are not obvious.
8. Are the comparisons in Fig 4D and 4E statistically significant?
9. Since there are three independent groups, one-way ANOVA (or non-parametric test deemed suitable) followed by permutation based post-hoc is likely more suitable for Figure 6B.
10. Line 348, the paper cited (Hedberg-Oldfors et al., 2018) did not comment on the possible cause of the craniofacial abnormalities is due to muscle weakness. There are other more appropriate citations.
11. Line 430, "there is a reduction in myotome size". It is unclear if this is true because while the DV length was shown to be reduced in Fig.4 but not the myotome volume. More specificity regarding the dimension of size that is impacted could be helpful.
12. Line 440, "... only partially involved in the larval and adult stages of muscle growth" is ambiguous. Mymx KO has significantly lower number of nuclei in adult fibers compared to WT. The use of words such as 'intermediate role' and 'partially involved' are unclear.
13. Discussion part should be strengthened with regard to the cellular requirements of Mymk and Mymx for cell-cell fusion. Authors should note that Tg(smyhc1:myomaker);myomixer-/- transgenic zebrafish (Yong et al., 2024, <https://doi.org/10.1016/j.jgg.2024.08.006>) have reported multinucleated fiber, that indicated the presence of Mymk alone in both fusing cells can lead to a low level of fusion and explained the less severe phenotype of Mymx KO.

Transfer to Biology Open

Author response to reviewers' comments

Reviewer comments

We thank the Reviewers for the comments on our manuscript. We are pleased they find the data presentation and quality of results generally high. We realise that the first version could have brought out the originality more clearly. We address this concern below, along with outlining changes made to the other suggestions (in blue). Changes in the manuscript are highlighted in red. In particular, we have improved the quantification of the skeleton results.

We hope with these changes that the manuscript can be accepted at Biology Open.

Before giving a detailed point-by-point rebuttal of the referee points, we address the issue of novelty. It is correct (and we cite extensively) that previous work has looked at *myomixer* mutants, including in the zebrafish. The previous studies showed that *Myomixer* is necessary for fusion and that the zebrafish mutants can survive to adulthood, although with impaired swimming. Here, we address a number of open questions:

1. What is the spatiotemporal pattern of *myomixer* expression in the early embryo, when the first wave of muscle fusion occurs?

As we cite (and noted by reviewers), previous work has done *in situs*, but these were: (i) low resolution; (ii) only focused on expression along the AP-axis; and (iii) qualitatively analysed, with no detailed exploration. As we showed in a previous study (Mendieta-Serrano *et al.* 2022), knowing the full four-dimensional (3 space + 1 time) expression pattern of *myomaker* was important in understanding the spatiotemporal occurrence of cell fusion. We perform a similar analysis here for *myomixer*.

2. How are future muscle cell dynamics impacted under loss of *myomixer*?

Previous work mostly relied on fixed embryos, for either *in situ* or antibody staining. However, the loss of fusion can impact more than just fusion itself, but also the cell movements (Mendieta-Serrano *et al.* 2022). Previous live imaging of *myomixer* embryos focused on demonstrating the absence of fusion, not on the cell dynamics or impact on tissue architecture. We have performed the first quantitative analysis of cell movement in live *myomixer* embryos, which revealed that the mutation impacts cell morphodynamics and alters the tissue architecture.

3. How does reduction in cell fusion impact adult form?

Previous work has shown that loss of *myomixer* leads to reduced muscle mass, with lower nuclei number, and impacted swimming. However, perturbations to muscle can impact the whole-body form, including the bone structure. We performed the first characterisation of the zebrafish skeleton under loss of *myomixer*. We show that the loss of *myomixer* leads to interesting phenotypes that are not simply somewhere between wildtype and *myomaker* mutants. In some cases, the phenotypes are similar to *myomaker*, such as the development of extra bone growth in the spine. In other cases, the *myomixer* mutant more closely resembles the wild type. This shows that skeletal defects are highly sensitive to the loss of muscle mass from impaired fusion. It will be exciting to explore this further to understand the underlying mechanisms driving these differences. This would then have potential to inform musculoskeletal diseases by revealing the more complex interplay between muscle and skeleton.

We are confident that this work represents a valuable addition to our understanding of how loss of *myomixer* impacts development, from the embryo to the adult. We have moved results which are more confirmatory (e.g., previous Figure 6) to the Supplementary Information.

Reviewer 1:

- 1.1. Although fibre counts are lacking for the *mymx* mutant, the observed mononucleated fibres indeed suggest a fusion deficit. Fast fibres resemble slow fibre in many aspects (single,

centralized myonuclei in fibres, fibre orientation) and it would be interesting to know whether these fibres also adopt a slow fate identity on the molecular level.

This is a very interesting question, but one not currently easily accessible. The molecular definition of slow vs fast fibres is not precisely defined. We have not seen expression of *Pdrm1a* in the mutants, a marker of slow cell fate, but this is not definitive. It would be a new study to properly dissect the nature of the muscle fibres under fusion inhibition. We have added a comment to the Discussion to expand on this (Lines 447-453).

- 1.2. Fig. 2: It remains untested in this study whether the mutant *Mymx* protein is lost, thus we recommend to change the title of Fig. 2 to "Loss of *mymx* function...". Please consider this caveat throughout the manuscript.

We have made these changes.

- 1.3. Fig. 3/4: These figures seem to represent the quantifications of data depicted in Fig. 2. It might make sense to pick the most telling results and combine these into one panel, while other results go into the supplementary material. Please insert "wt" and "*mymx*^{-/-}" into panel A of Fig. 3.

In this study, we dissect the *mymx*^{-/-} mutant, including careful quantification of myocyte behaviour. We would like to retain Figures 2-4 as they reveal distinct points: Figure 2 - outlines the mutation and shows the raw data that reveals changes in morphology at a tissue-scale; Figure 3 - this provides a quantification of the cell morphology from 30-120 hpf; Figure 4 - this figure provides quantification of the tissue morphology. We have updated the figures with genotypes.

- 1.4. Please list the previously established *mymk* mutant and transgenic lines (H2B-mCherry, lyn-GFP) with adequate allele designation in the methods section under "Zebrafish husbandry". Please follow the zebrafish nomenclature convention throughout text and figures (gene=*mymx*, protein=*Mymx*). See here: <https://zfin.atlassian.net/wiki/spaces/general/pages/1818394635/ZFIN+Zebrafish+Nomenclature+Conventions>

We have made these changes as requested.

Reviewer 2:

- 2.1. I am puzzled by the author's claimed "gap" in knowledge, that the expression and function of *Mymx* are not well understood. The authors state this supposed gap both in the abstract (lines 20-22) and in the introduction (lines 92-93). To substantiate this claim, they will need more clarity on what remains unknown about *mymx*.

We have changed the abstract and the introduction to bring out more clearly the novel aspects of this work (lines 102-111). See detailed discussion above.

- 2.2. The requirement for *mymx* in muscle cell fusion was known in zebrafish at least as far back as 2017, with a starting paper from Eric Olsen's lab with further work coming from Elizabeth Chen and Shaojun Du's groups. The current author's subsequent claims of novelty also run into trouble. Wu 2022 showed that *mymx* is required for growth to normal adult sizes and for myofiber growth too. The only part of the authors conclusions that seems potentially new is the impact of *mymx* on skeleton, which is not explored quantitatively in the current work.

See our above comment about novelty. We have now expanded the quantification of the skeletal defects, as suggested (new Figure 7) and lines 389-397.

2.3. Major comments:

- 2.4. The authors provide a brief exploration of *mymx* gene expression patterns using fluorescent ISH instead of the previously published DIG ISH. This is not a comprehensive examination and only explores an N=3 animals at a single stage of development (24 hpf). The authors did assess

expression across a multitude of ROI at this stage, but that does not change the fact that there are three animals, one stage, and not many believable surprises.

This result was to define clearly the expression pattern. A key advantage of zebrafish is that we can image many somite stages at once. Though we take one time point, within this each somite is at a different developmental time. Anterior somites are “older”, with each subsequent somite being around 30 minutes “younger”. Therefore, by analysing multiple somites, we are building a picture of the temporal pattern of *myomaker* expression. We clarify the time point studied. The timepoint chosen also overlapped with high levels of cell fusion (Mendieta-Serrano *et al.* 2022).

Regarding the n, we analysed at least five segments per embryo, so we are extracting a lot of information from a single fish. The patterns were highly reproducible between the three embryos. Further, the results are consistent (in the AP-axis) with previously published work. Therefore, we are confident the data presented here faithfully follows the mean behaviour of *myomixer* expression pattern. We have added a note in the Discussion about exploring further fusion events after the primary wave of fusion (lines 514-518).

2.5. They provide abundant quantification of the number of myonuclei found in different somite levels in *mymx* through the first week of life. (The WT fusion levels have already been shown in other studies). This reveals that muscle cells occasionally fuse in *mymx*^{-/-}. Rare fusion events like these in *mymx*^{-/-} were previously demonstrated in Wu *et al.*, 2022 "Loss of Myomixer Results in Defective Myoblast Fusion, Impaired Muscle Growth, and Severe Myopathy in Zebrafish"

Wu *et al.* have shown that fusion still persists with loss of *Mymx* function and that there are changes in muscle structure (e.g., Figure 9 from that work). The results shown here are focused on comparing the specific differences between wild type, *mymx*^{-/-} and *mymk*^{-/-} mutants. This more clearly highlights that the phenotype in *mymx*^{-/-} mutants lies between wild-type and *mymk*^{-/-} in terms of severity. While we believe this comparison is interesting, this is similar to an observation that has been made previously. Therefore, we have moved Figure 6 to the SI.

2.6. The authors show that *mymx* mutants have somites that are briefly taller than WT, but they do not make clear why this transient finding matters.

An important difference between our work and previous studies is that we analysed not just fusion (or lack thereof) but also the impact of this on the zebrafish myotome morphology (previous work has only shown that muscle cell size is impacted by loss of fusion). The observation that loss of fusion impacts somite shape is interesting, as it shows that loss of fusion at a cell-scale can impact tissue-scale behaviour (note, we still have the same number of nuclei between *myomixer* mutants and wild type). Of course, the question whether it matters is a challenging one to address. We show in (new) Figure 6-7 that loss of fusion impacts formation of the skeleton. Is this, in part, due to earlier changes in somite morphology, even if only transient? Such questions are interesting but are suitable for further study. We have updated the Discussion to highlight this (lines 494-497).

2.7. The authors provide stills from a time-lapse of cell fusion in the *mymx* mutant. In these stills one can see that the mutant does not fuse, which already well known. Furthermore, a higher-resolution time-lapse of muscle cell fusion in another *mymx* mutant was provided previously by Luo *et al.*, 2022. In an attempt to make new insights, the authors provide new quantification (panel B) in which the error bars between WT and *mymx*^{-/-} overlap at every timepoint, they then zoom in on a region where the two time-lapse videos might differ and remove the error bars. The lack of error bars in this inset borders on deceptive.

We highlight here that we are interested in the cell dynamics, not just the absence of fusion. This was motivated by observations from our previous and recent work (Mendieta-Serrano *et al.* 2022; Mendieta-Serrano *et al.* *Biorxiv* 2025), where *myomaker* mutants displayed interesting cellular dynamics. The movie from Luo *et al.*, is focused on the formation of actin puncta. They do not provide cell tracking, and do not focus on how the tissue architecture is altered - their research question was distinct from ours.

We did not show error bars on the inset for clarity - they are clearly exhibited in the main panel and there was no intention to deceive. The difference in the average speed between the wild type and *myomixer* mutant from 500-700 minutes is statistically different. We have now added this to the paper and clarified the figure results (lines 321-323).

2.8. The authors provide examples of isolated myofibers from adult *mymx* mutants (presumably fast-twitch ones). They then show that *mymx*^{-/-} mutants have a strong, but partial fusion defect that is less severe than *mymk*^{-/-}. However, this finding has also been shown previously with high similarity to Wu 2022, Figure 9.

We agree with the reviewer that such a phenotype has been previously shown. The original Figure 6 was presented to give context to Figure 7. We have now moved this figure to the Supplementary Material. We do note that it is beneficial to have confirmatory results, as this gives us confidence in our approach.

2.9. Finally, in Figure 7, the authors make an arguably new claim. They provide a non-quantitative exploration of adult skeletal defects in the fusion deficient *mymk*^{-/-} and *mymx*^{-/-} mutants, with no mechanistic insights or tests.

Repeating point 3 from our initial statement on novelty.

Previous work has shown that loss of *myomixer* leads to reduced muscle mass, with lower nuclei number, and impacted swimming. However, perturbations to muscle can impact the whole-body form, including the bone structure. We performed the first characterisation of the zebrafish skeleton under loss of *myomixer*. We show that the loss of *myomixer* leads to interesting phenotypes that are not simply somewhere between wildtype and *myomaker* mutants. In some cases, the phenotypes are similar to loss of *myomaker*, such as the development of extra bone growth in the spine. In other cases, the *myomixer* mutant more resembles the wild type. This shows that skeletal defects are highly sensitive to the loss of muscle mass due to impaired fusion. It will be exciting to explore this further, to understand the underlying mechanisms driving these differences. In particular, when do such skeletal defects emerge? Is it early in bone development or later once the fry are swimming freely? This would then have potential to inform musculoskeletal diseases by revealing the more complex interplay between muscle and skeleton. However, such a detailed mechanistic analysis is a study in its own right.

Reviewer 3:

- 3.1 Overall, the manuscript is descriptive and lacks new mechanistic insights on how loss of *Mymx* causes secondary cellular defects in addition to perturbed fusion. Some phenotypic analysis was detailly executed such as myotome angle, myotome volume and adult myonuclei number which has shed light on the essential role of *Mymx* during myogenesis. However, the rationale of analyzing the cell rearrangement dynamics in *Mymx* mutant is not clear. Also, *Mymx* KO mutant in this manuscript has a largely overlapping phenotype with the previously reported mutants, thus overall there is unclear new knowledge that would be valuable for the field.
- 3.2 Below are specific comments about the current data presented, but without a better narrative with more obvious advances for the field, I do not think this manuscript is optimal for Development.

See our statement above and we have improved the Introduction to outline the rationale more clearly. We have also extended the Discussion to bring out interesting future directions.

- 3.3 The abstract says that *Mymx* function is less well understood. I think it is incorrect since there are papers specifically deciphering *Mymx* activity but these are lacking for *Mymk*.

We have reworded.

3.4 Line 138: "mono-nucleated slow-twitch muscles that do not express *mymk* or *mymx*" may be inaccurate. It was reported *Mymk* transcripts are detected transiently in slow muscle precursors adaxial cells (Yong, 2024).

We have reworded (lines 144-147) and further added to the Discussion (lines 509-511).

3.5 In Fig. 1, *Mymx* expression along AP axis is only shown for the 24-hpf but Fig. 1D shows the fluorescent intensity across 5 timepoints. As reported in Shi. et. al, 2017, 24-hpf is not the timepoint with highest *Mymx* expression. Should authors show all *Mymx* staining along AP axis to complement the signal intensity quantification? Following that, can the authors describe the *Mymx* expression pattern and highlight the relevance?

We chose this time point as it associates well with when fusion events are common in these segments (Mendieta-Serrano *et al.* 2022) (lines 121-122).

3.6 Line 148, "*mymx* has a complicated pattern of expression" is ambiguous. It would be great if the authors could explain the potential relevance, if any, of this expression pattern.

We have improved this discussion (lines 156-159) and included relevance to *myomaker* and the observed waves of fusion (Mendieta-Serrano *et al.* 2022).

3.7 Line 200, "low level of myocyte fusion can occur..." based on the presence of binucleated cells. Are binucleated cells definitive or could this be cells that are undergoing cell division?

This is a good question. We do observe cell divisions in the somite, but these are rare (typically only a few per segment, Mendieta-Serrano *et al.* 2022). The morphology of a dividing cell is very distinct from typical muscle fibres. Therefore, we're confident that we are not conflating divisions and fusion events. We have added a point on this, lines 218-221.

3.8 Since the authors have already shown nuclei number per myofiber, it is unclear what the analysis of number of nuclei per myotome segment (Figure 4E) is achieving.

These results were placed to summarise our observations. We have now moved to the Supplementary information.

3.9 Line 249-252, it is unclear how a smaller dorsoventral length and similar overall volume represent a lower capacity for muscle hypertrophy and reduced expansion in size of the muscle fibers. It is confusing because the authors say that overall volume is similar but there is reduced expansion in size. Overall, the concepts to be conveyed are not obvious.

We have reworked (lines (267-271)).

3.10 Are the comparisons in Fig 4D and 4E statistically significant?

We have now included such tests - note, these have been moved to supplementary information.

3.11 Since there are three independent groups, one-way ANOVA (or non-parametric test deemed suitable) followed by permutation based post-hoc is likely more suitable for Figure 6B.

We have utilised estimation statistics to perform the analysis (Ho *et al.* Nature Methods 2019) and also moved to the Supplementary information.

3.12 Line 348, the paper cited (Hedberg-Oldfors *et al.*, 2018) did not comment on the possibly cause of the craniofacial abnormalities is due to muscle weakness. There are other more appropriate citations.

We have updated.

- 3.13 Line 430, "there is a reduction in myotome size". It is unclear if this is true because while the DV length was shown to be reduced in Fig.4 but not the myotome volume. More specificity regarding the dimension of size that is impacted could be helpful.

We have clarified (lines 467-468).

- 3.14 Line 440, "... only partially involved in the larval and adult stages of muscle growth" is ambiguous. Mymx KO has significantly lower number of nuclei in adult fibers compared to WT. The use of words such as 'intermediate role' and 'partially involved' are unclear.

We have reworded (lines 479-480).

- 3.15 Discussion part should be strengthened with regard to the cellular requirements of Mymk and Mymx for cell-cell fusion. Authors should note that Tg(smyhc1:myomaker);myomixer-/- transgenic zebrafish (Yong et al., 2024, <https://doi.org/10.1016/j.jgg.2024.08.006>) have reported multinucleated fiber, that indicated the presence of Mymk alone in both fusing cells can lead to a low level of fusion and explained the less severe phenotype of Mymx KO.

We have now included (lines 509-511).

First decision letter

MS ID#: bio.062305

MS Title: Dynamic expression and differential requirement of the myocyte fusogen Myomixer during distinct myogenic episodes in the zebrafish

Authors: Sunandan Dhar; Serena Thomas; Hui Li Yeo; Timothy Saunders; Sudipto Roy
Article Type: Transferred Research Article

Dear Dr Saunders,

I am happy to tell you that your manuscript has been accepted for publication in Biology Open, pending our standard publication integrity checks. It was accepted on 18th November 2025.